# Exploring Dengue Infection in a Vaccinated Individual: Preliminary Molecular Diagnosis and Sequencing Insights

**DOI:** 10.3390/v16101603

**Published:** 2024-10-12

**Authors:** Talita Émile Ribeiro Adelino, Sílvia Helena Sousa Pietra Pedroso, Maurício Lima, Luiz Marcelo Ribeiro Tomé, Natália Rocha Guimarães, Vagner Fonseca, Paulo Eduardo de Souza da Silva, Keldenn Melo Farias Moreno, Ana Cândida Araújo e Silva, Náthale Rodrigues Pinheiro, Carolina Senra Alves de Souza, Luiz Carlos Junior Alcantara, Marta Giovanetti, Felipe Campos de Melo Iani

**Affiliations:** 1Fundação Ezequiel Dias, Belo Horizonte 30510-010, Brazil; shspietra@gmail.com (S.H.S.P.P.); maurili15@gmail.com (M.L.); lmarcelotome@gmail.com (L.M.R.T.); natyroguiman@gmail.com (N.R.G.); pauloeduss@gmail.com (P.E.d.S.d.S.); keldennmfmoreno@gmail.com (K.M.F.M.); 2Instituto René Rachou, Fundação Oswaldo Cruz, Belo Horizonte 30190-009, Brazil; alcantaraluiz42@gmail.com; 3Department of Exact and Earth Science, University of the State of Bahia, Salvador 41192-010, Brazil; vagner.fonseca@gmail.com; 4Climate Amplified Diseases and Epidemics (CLIMADE), Brasília 70070-130, Brazil; 5Centre for Epidemic Response and Innovation (CERI), School of Data Science and Computational Thinking, Stellenbosch University, Stellenbosch 7600, South Africa; 6Instituto de Ciências Biológicas, Universidade Federal de Minas Gerais, Belo Horizonte 31270-901, Brazil; 7Comitê Técnico Científico Multidisciplinar, Universidade Federal dos Vales do Jequitinhonha e Mucuri, Teófilo Otoni 39800-091, Brazil; ana.candida@ufvjm.edu.br (A.C.A.e.S.); nathale.pinheiro@ufvjm.edu.br (N.R.P.); 8Secretaria Estadual de Saúde de Minas Gerais, Belo Horizonte 31630-900, Brazil; carol.senraas@gmail.com; 9Sciences and Technologies for Sustainable Development and One Health, Università Campus Bio-Medico di Roma, 00128 Roma, Italy

**Keywords:** DENV monitoring, next-generation sequencing, genomic surveillance, vaccine breakthrough

## Abstract

This study examines a case involving a 7-year-old child who developed dengue symptoms following Qdenga vaccination. Despite initial negative diagnostic results, molecular analysis confirmed an infection with DENV4. Next-generation sequencing detected viral RNA from both DENV2 and DENV4 serotypes, which were identified as vaccine-derived strains using specific primers. Phylogenetic analysis further confirmed that these sequences belonged to the Qdenga vaccine rather than circulating wild-type viruses. This case underscores the critical need for precise diagnostic interpretation in vaccinated individuals to avoid misdiagnosis and to strengthen public health surveillance. A comprehensive understanding of vaccine-induced viremia is essential for refining dengue surveillance, improving diagnostic accuracy, and informing public health strategies in endemic regions.

## 1. Introduction

Dengue fever, caused by dengue viruses (*Orthoflavivirus denguei*, DENV), is a mosquito-borne viral disease that poses a significant public health concern worldwide, especially in tropical and subtropical regions [1]. DENV, a single-stranded RNA virus, comprises four distinct serotypes (DENV1 to DENV4), exhibiting unique genetic and antigenic properties that contribute to a wide spectrum of clinical manifestations [2,3]. Since its introduction in Brazil in the 1980s, dengue has presented a major epidemiological challenge. In 2024, Brazil reported 6,486,906 probable cases of dengue, with over 1,696,426 cases documented in the state of Minas Gerais alone as of August 2024, making it the second highest incidence of the disease [4]. This year, Minas Gerais, located in southeastern Brazil, has experienced the co-circulation of DENV1, DENV2, and DENV3 serotypes, further complicating the local epidemiological landscape [5].

Recent advances in dengue prevention include the introduction of the Qdenga vaccine (TAK-003 by Takeda Pharmaceuticals) into the Brazilian public healthcare system. Qdenga is a live attenuated tetravalent vaccine derived from a DENV2 backbone incorporated with recombinant strains expressing surface proteins for DENV1, DENV3, and DENV4. Clinical trials have demonstrated its efficacy in reducing both the incidence and severity of dengue fever [6], offering promising prospects for mitigating dengue-related morbidity and mortality in endemic regions. However, the vaccine leaflet warns of side effects observed during studies in children, adolescents, and adults [7]. Commonly reported reactions include pain at the injection site, headache, muscle pain, general discomfort, weakness, respiratory infections, fever, irritability, and drowsiness. Less frequent adverse effects, occurring in 0.1% to 1% of patients, include gastrointestinal symptoms such as diarrhea, nausea, and vomiting, as well as dizziness, skin itching, hives, and fatigue. Rarely, up to 0.01% of patients experience rapid swelling under the skin, particularly on the face, throat, arms, and legs [8,9]. Additionally, transient viremia following Qdenga vaccination was observed in 49% of study participants who had no prior dengue infection and in 16% of those with a history of dengue infection [10]. This scenario is particularly concerning for surveillance purposes, as dengue diagnostic tests may yield positive results during vaccine-induced viremia, potentially leading to misinterpretation as the emergence of serotypes or genotypes not currently circulating in the region.

Understanding dengue-vaccine-associated infections is essential for preventing misdiagnoses and ensuring the effectiveness of surveillance programs globally. In this study, we provide a detailed characterization of the first Qdenga vaccine-associated case in the state of Minas Gerais, southeast Brazil, underscoring the urgent need for accurate monitoring to protect public health.

## 2. Materials and Methods

### 2.1. Sample Collection and Molecular Diagnostic Assay

A serum sample from a 7-year-old male suspected of DENV4 infection was submitted to the Central Laboratory of Public Health of Minas Gerais (Belo Horizonte, Brazil) at the Fundação Ezequiel Dias (Funed) for diagnostic confirmation. Viral RNA extraction was performed using an automated protocol on the Extracta 96 system (Loccus, Cutia, Brazil) with the Extracta Kit DNA and RNA Pathogen MDx (Loccus), following the manufacturer’s instructions. Molecular diagnosis was conducted on the CFX-96 Real Time System (Bio-Rad, Hercules, CA, USA) using the Molecular ZDC-IBMP Kit, developed by the Instituto de Biologia Molecular do Paraná (Santa Catarina, Brazil) and provided by the Brazilian Ministry of Health (BrMoH) to the public laboratory network. This kit enables detection and differential diagnosis of arboviruses through RT-qPCR multiplex reactions for DENV serotyping (DENV1–4), Zika virus (*Orthoflavivirus zikaense*), chikungunya virus (*Alphavirus chikungunya*), and the human RNAseP gene, serving as an endogenous control.

### 2.2. cDNA Synthesis and Whole-Genome Sequencing Using Nanopore Technology

Viral RNA was subjected to cDNA synthesis using the ProtoScript II First Strand cDNA Synthesis Kit (NEB—Ipswich, MA, USA), following the manufacturer’s instructions. The resulting cDNA was then submitted to sequencing multiplex PCR (35 cycles) using Q5 High Fidelity Hot-Start DNA Polymerase (NEB—Ipswich, MA, USA) and a set of specific primers designed by the CADDE project (https://www.caddecentre.org/, accessed on 20 March 2022) to set the complete genomes of DENV1 to DENV4 serotypes. Amplicons were purified using 1x AMPure XP Beads (Beckman Coulter—Brea, CA, USA) and quantified with a Qubit 3.0 fluorimeter (Thermofisher Scientific, Waltham, MA, USA) using the Qubit™ dsDNA HS Assay Kit (Thermofisher Scientific, Waltham, MA, USA). DNA library preparation was performed with the Ligation Sequencing Kit (SQK-LSK109) and the Native Barcoding Kit (EXP-NBD104) from Oxford Nanopore Technologies (ONT) (Oxford, UK) [11], followed by sequencing on an R9.4 flow cell using the MinION platform (ONT) (Oxford, UK).

### 2.3. Generation of Consensus Sequence

Raw files were basecalled and demultiplexed using Guppy v.6.0 (Oxford Nanopore Technologies). Consensus sequences were generated by a hybrid approach using the Genome Detective online tool (https://www.genomedetective.com/, accessed on 12 February 2024) [12]. DENV genotyping was assessed using the Dengue Virus Typing Tool (https://www.genomedetective.com/app/typingtool/dengue/, accessed on 12 February 2024) [13], integrated within the Genome Detective platform. The newly generated DENV2 and DENV4 sequences were deposited in GISAID under accession numbers EPI_ISL_19365344 and EPI_ISL_19365345, respectively.

### 2.4. Phylogenetic Analysis

To determine the origin of the DENV2 and DENV4 genomes from this case, we performed maximum likelihood (ML) phylogenetic analyses using two distinct datasets. The DENV2 dataset was constructed with our new DENV2 sequence, along with 292 reference DENV2 complete genomes representing various genotypes deposited in public database (GenBank and GISAID) (Genotype I n = 20, Genotype II n = 54, Genotype III n = 89, Genotype IV n = 35, Genotype V n = 35, and Genotype VI n = 3). This dataset also included the DENV2 genomes isolated from three Qdenga vaccine-associated cases in the Brazilian state of Mato Grosso do Sul (EPI_ISL_18877651, EPI_ISL_19115812, and EPI_ISL_19115813). Additionally, our dataset included the sequence KU725663, which is the vaccine strain used by Takeda. Similarly, the DENV4 dataset comprised our new DENV4 sequence and 476 sequences from the prM/E region, representing various genotypes (Genotype I n = 191, Genotype II n = 277, and Genotype III n = 8). This dataset also included the DENV4 sequence isolated from a Qdenga vaccine-associated case in Mato Grosso do Sul (EPI_ISL_18877652). Notably, our dataset also included the sequence MW793460, which is the vaccine strain used by Takeda.

Sequence alignment for each dataset was performed using MAFFT v7.3.10 [14] and manually curated to remove artifacts using AliView v1.28 [15]. ML phylogenetic trees were estimated using IQTREE v2.3.6 [16] applying the best-fit model inferred by the ModelFinder application implemented within IQTREE. Branch support was assessed using the approximate likelihood-ratio test incorporating bootstrap and the Shimodaira–Hasegawa-like procedure with 1000 replicates.

## 3. Results

On 29 January 2024, a 7-year-old male child received a dose of the Qdenga vaccine. Six days later, on 4 February 2024, he developed symptoms including fever, myalgia, rash, and sore throat. On 6 February 2024, the patient was admitted to a public health unit in the city of Teófilo Otoni, Minas Gerais, Southeast Brazil. There, he underwent a dengue NS1 rapid test, yielding a negative result. On the same day, a serum sample was collected and sent to the Universidade Federal dos Vales do Jequitinhonha e Mucuri (UFVJM) campus in Teófilo Otoni, for molecular diagnosis of arboviruses. This institution is part of the accredited laboratory network of the Secretaria de Estado de Saúde de Minas Gerais (SES-MG). The RT-qPCR assay using the Molecular ZDC-IBMP Kit detected the DENV4 serotype, with a cycle threshold (CT) value of 29. Due to the epidemiological significance of this case, particularly because this serotype is not currently circulating in Brazil and the molecular diagnostic kit used does not differentiate between the wild-type virus and the vaccine strains from Qdenga, the sample was sent to Lacen-MG, located at Funed, for confirmation and genetic characterization through next-generation sequencing.

At Funed, the sample underwent further molecular diagnosis using the Molecular ZDC-IBMP Kit, which confirmed the detection of the DENV4 serotype, this time with a CT value of 32. To further investigate the possibility of a Qdenga vaccine-associated case, next-generation sequencing was performed using the MinION platform (ONT) and four specific primer sets designed to cover the entire coding region of each of the four DENV serotypes, as outlined in previous protocols (https://www.caddecentre.org/protocols/, accessed on 20 March 2022).

The sequencing procedures revealed the presence of both DENV2 and DENV4 serotypes. For the DENV2, a total of 484,441 reads were obtained, with an average depth of 49,563× and a coverage of 59.7%. In contrast, the DENV4 genome yielded 256,966 reads, with an average depth of 169.9× and a coverage of 9.3%. Although we used specific primer sets designed to cover the entire coding region of each DENV serotype, the DENV4 reads identified corresponded only to a fragment of the prM/E region of this serotype, similar to the insert used in the Qdenga vaccine. Using the online Dengue Typing Tool, the DENV2 and DENV4 sequences were classified as DENV2 genotype V (DENV2-V), which was not recorded as circulating in Brazil, and DENV4 genotype II (DENV4-II), respectively. Phylogenetic analysis subsequently confirmed the genotypes identified previously by the online tool (Figure 1).

In the phylogenetic tree, the DENV2 genome obtained from Minas Gerais clustered with other Brazilian strains sampled in 2024 (EPI_ISL_19115812, EPI_ISL_19115813, and EPI_ISL_18877651). These sequences were isolated from suspected cases of Adverse Event Following Immunization (AEFI) in the Brazilian state of Mato Grosso do Sul, where patients had recently received the Qdenga vaccine. Additionally, this clade included the DENV2 genome strain 16681 (Mahidol strain, KU725663), which was the original strain used as backbone for Takeda’s Qdenga vaccine, and, after attenuation, became known as DENV2 PDK-53 [17,18] (Figure 1).

Regarding the DENV4 serotype, the newly generated sequence from Minas Gerais clustered with another Brazilian strain sampled in 2024 (EPI_ISL_18877652), isolated from a suspected AEFI case in Mato Grosso do Sul state. This clade also included the sequences MW793460 and KX812530, corresponding to DENV4 isolate 1036 (Figure 1), which was used in the production of the Qdenga vaccine [17,19].

## 4. Discussion

In this study, we describe the probable Qdenga vaccine-associated case in the state of Minas Gerais, southeastern Brazil, involving a male child from Teófilo Otoni, as indicated by sequencing results and phylogenetic analyses. Therefore, the purpose of this case report is to alert public and private laboratories about the importance of accurately interpreting diagnoses of DENV serotypes that are not currently circulating in a country. The introduction of a new DENV serotype could result in an increase in dengue cases and severity of clinical symptoms [20,21,22]. This highlights the critical need for accurate monitoring to protect public health and ensure the effectiveness of public policies.

Phylogenetic analyses revealed that the newly sequenced DENV2 genome belongs to genotype V. Since its initial detection in Brazil in the early 1990s [23], DENV2 have been responsible for widespread epidemics [22]. Genotype III, also known as the Southeast Asian-American, was the predominant genotype in Brazil until 2021, when genotype II, also known as Cosmopolitan, was first detected in the country [24]. In contrast, genotype V is mostly confined to Asia [25], with no evidence of autochthonous circulation in Brazil to date. Our new DENV2-V strain from Minas Gerais clustered with other Brazilian isolates from suspected cases of AEFI and with the DENV2 genome strain 16681, which, after attenuation, was used as the backbone for the Qdenga vaccine [17,18]. The sequencing results also revealed the presence of the DENV4 serotype, specifically corresponding to the prM/E region, similar to the insert used in the Qdenga vaccine [8]. Taken together, these findings suggest that this case is the result of transient viremia following the administration of the Qdenga vaccine.

Vaccine-induced viremia, as observed in this case report, is a phenomenon seen with other vaccines, such as those for Rubella and Yellow Fever [26,27]. This occurs because these vaccines contain attenuated or inactivated forms of the virus that can temporarily replicate in the body to stimulate a humoral immune response, especially a cellular response mediated by CD8+ lymphocytes as mediators of protection [28]. Importantly, while this phenomenon may occur, it is primarily influenced by the immune status of the recipient at the time of vaccination, rather than being an inherent property of the vaccine, although the potential exists. Temporary viremia is usually short-lived and does not cause illness, but it is sufficient to induce the production of antibodies and provide immunity against the actual infection [29]. The primary mechanism of action of the Qdenga vaccine is to replicate locally and elicit neutralizing antibodies to confer protection against dengue caused by any of the four DENV serotypes. The vaccine activates multiple arms of the immune system, including binding antibodies, complement-fixing antibodies, functional antibodies to dengue NS1, and cell-mediated immune responses [9]. During clinical trials of the Qdenga vaccine, efficacy varied by individual serotypes (DENV 1, 69.8%; DENV 2, 95.1%; DENV 3, 48.9%; DENV 4, 51%). Cumulative rates of serious adverse events were similar in experimental group (4%) and placebo (4.8%) recipients and were consistent with expected medical disorders in the study population. Among those who presented with a febrile illness within 30 days of vaccination, vaccine viraemia was detected in 7% of participants after the first dose. The Qdenga vaccine was well tolerated and efficacious against symptomatic dengue regardless of serostatus before immunization. Thus, it was deemed safe for administration by regulatory agencies [29].

The side effects observed in this report were not different from those seen during the clinical testing phase of the vaccine, and the patient described did not experience any severe effects. However, caution must be taken when interpreting positive diagnostic tests for non-circulating strains within the country, especially in post-vaccination patients. A thorough patient history should be conducted, and public health laboratories that perform genomic characterization to differentiate between natural infection cases and those associated with vaccination should work towards better determining the clinical case before notifying epidemiological authorities.

In summary, our results suggest that the identification of both DENV2 and DENV4 serotypes in the analyzed sample is not related to the recent emergence of wild-type strains, but rather to the presence of a genetically modified vaccine clone of the Qdenga vaccine. This clone consists of a DENV2 backbone and expresses the prM/E region of the DENV4 serotype, which has the highest titer in the vaccine (≥4.5 log10 PFU/dose) [7]. In this context, understanding dengue-vaccine-associated infections, including the immune status of the patients, informs public health strategies, emphasizing the importance of surveillance, rapid diagnosis, and genomic characterization. Such insights are crucial for developing more effective preventive measures and enhancing vaccine safety, thereby mitigating the impact of dengue.

## Figures and Tables

**Figure 1 viruses-16-01603-f001:**
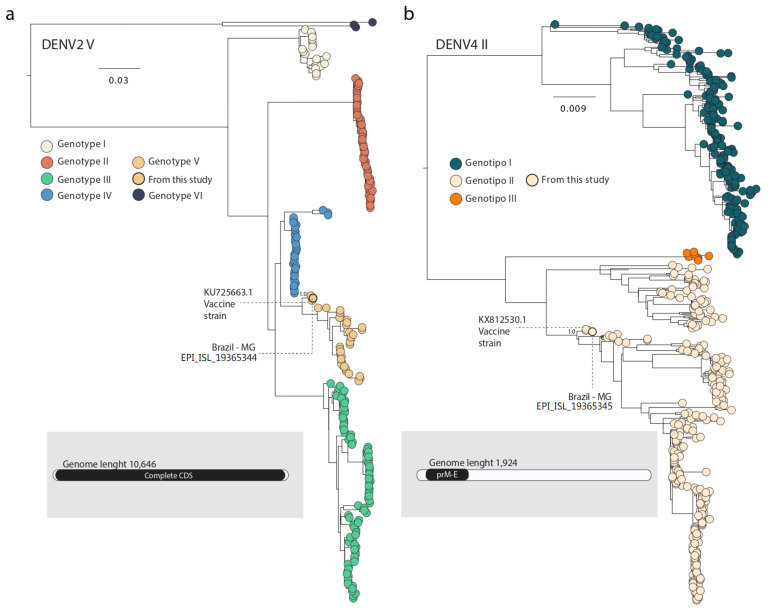
Phylogenetic reconstruction of DENV2 and DENV4. (**a**) Midpoint rooted maximum-likelihood phylogeny of distinct DENV2 genotypes, highlighting our newly generated DENV2-V sequence from Minas Gerais (EPI_ISL_19365344) and the Takeda vaccine-strain KU725663. The inset at the bottom left of panel shows the genome coverage retrieved from the pathogen, with a total genome length of 10,646 base pairs, covering the complete CDS (coding sequence) of the virus. (**b**) Midpoint rooted maximum-likelihood phylogeny of distinct DENV4 genotypes, highlighting our newly generated DENV4-II sequence from Minas Gerais (EPI_ISL_19365345) and the Takeda vaccine-strain KX812530.The inset at the bottom right of panel represents the genome coverage for the DENV4 sequence retrieved from the pathogen, displaying a genome length of 1924 base pairs, specifically covering the prM/E region of the genome.

## Data Availability

Newly generated DENV2 and DENV4 sequences have been deposited in GISAID under accession numbers EPI_ISL_19365344 and EPI_ISL_19365345.

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
