# Peer review of "Exploring Dengue Infection in a Vaccinated Individual: Preliminary Molecular Diagnosis and Sequencing Insights"

_viruses, 2024, doi:10.3390/v16101603_

Round 1

Reviewer 1 Report

Comments and Suggestions for Authors

Exploring dengue infection in a vaccinated individual: preliminary molecular diagnosis and sequencing insights

Adelino et al.

In this short report, the authors describe a case of Qdenga vaccine associated dengue symptoms. A 7-year-old male child developed dengue symptoms following Qdenga vaccination. Although initial NS1 rapid test was negative, subsequent RT-PCR and sequencing results identified the presence of DENV2 and DENV4 serotypes, which originated from the vaccine strains. These results raise concerns about how to distinguish between symptoms caused by real infections and those associated with the vaccine. The manuscript is fairly well written. However, I wonder why the authors did not attempt to culture the samples to isolate the viruses. Additionally, it is unclear whether the symptoms were linked to any other virus infections or whether tests were conducted for other arbovirus infections. Section 3 should be titled “Results” rather than “Case Description.”

Minor point

Line 152, At Funed, the sample (missing) underwent further molecular diagnosis …  

Reviewer 2 Report

Comments and Suggestions for Authors

In this manuscript, the authors explore presence of infection in Qdenga, and suggest how these findings can potentially affect interpretation of infection outcome. While I understand this may be relevant in the context of surveillance, I also want to highlight that this is an expected outcome because live-attenuated vaccines are known to cause transient viremia. For instance, the yellow fever live-attenuated vaccine can exhibit viremia for up to 7-14 days but do not cause much symptoms as the virus is attenuated.

In the case study, it is also unknown when the viruses were assessed after vaccination. Was the virus detected a few days after vaccination, or did they persist long-term in the individual? It is also unknown how the kinetics of the viremia is, which is critical to interpret the viral load and the symptomatic outcome of the individual. How much of the viral load was detected, and were they at levels that may explain for symptom outcome? Also, Qdenga is a tetravalent vaccine. Were the other serotypes undetected?

Reviewer 3 Report

Comments and Suggestions for Authors

Dear Authors,

I have some comments and recommendations to implement in order to get your manuscript "Exploring Dengue Infection in a Vaccinated Individual: Preliminary Molecular Diagnosis and Sequencing Insights" (which is a Case Report) ready for publication:

1) In lines 91-93 you wrote “ This kit enables detection and differential

diagnosis of arboviruses through RT-qPCR multiplex reactions for DENV serotyping (DENV1-4),…” . Could you please note that it depends on which RNA it detects. Because if it is from the pre-membrane (prM) or envelope (E) genes you won’t be able to distinguish between a WT infection and the vaccine attenuated viral version. 

2) In line 127 you make clear that your DENV4 dataset comprises 476 sequences from prM/E regions...Just as I say before, these genes are exactly the ones you cannot pick up to say that this strain is not a wild type natural infection and belongs to a vaccine derived one, because you will not be able to distinguish is the amplified zone is from the vaccine, mutated afterwards or during the infection or recombined somehow with other sequences... the only way to ascertain that the virus comes from the vaccine is that you try to amplify from the same samples non structural proteins genes from DENV4, which is not in the vaccine. If it is negative for NSs, but positive for prM/E DenV4 genes, then you can assume that it comes from the vaccine. Otherwise it is not enough evidence to say it comes from the vaccine.

3) Lines 160 to 162, I have the same concern here. The unaware reader will take your data as the revealed truth if you do not explain that you don't know IF THIS IS FROM THE VACCINE OR A WILD TYPE virus.

4) Lines 194-196 You should add the word "possible" because you did not sequence the Denv 4 virus for NS genes, if you would do so, you will have proof that it is vaccine related only (if no DENV 4 NS sequence was obtained in all the amplicons).

5) Lines 200-201. I believe you should tone down your claims, because you are talking about one case in how many vaccinees? 1/10.000? 1/100.000? And besides this, you do not have the smoking-gun proof that it is a reversion from the vaccine. Moreover, did anybody measured the immune status of that child? What about that? Please tone down your claims.

6) Lines 216 to 218. Maybe… But importantly, this occurs also because a weak immune system of the recipient (etc,) at the moment of vaccination, because in most vaccinees this did not occur, so it is mostly a property of the recipient, and not of the vaccine itself (although the potentiality exist).Please make it clear in the text.

7) Lines 220 to 222. Antibodies alone does nothing (exactly like antibody-dependent enhancement -ADE- in Dengue -and probably also in SARS-CoV-), the main goal of live attenuated is that the replication inside a cell and a killing inside a cell generates that viral proteins are degraded in proteasome and cross-presented to CD8-T cells, that rapidly kill infected cells to avoid a viral population explosion in the next cycle of replication. PLEASE INCLUDE THIS IN THE MANUSCRIPT (reformulate to something better but maintaining the concept).

8) Lines 224-225. THis is the only feature of live attenuated that you will not achieve with other vaccine types, this is why they are more effective, beside maybe better antibody affinity or titters/ class switch earlier, etc (this is just a comment, you do not need to place it in the manuscript)

9) Lines 236-237. Maybe, the recommendation should be to genotype NS proteins to be sure it is not vaccine related, at least in cases of DENV4.

10) Lines 245 to 249. For these very rare events, may be useful to have a genomic characterization protocol, including DENV4 NS genes, and it will be very important to know the immune status of the patient at the time of vaccination or just before the unwanted reactions. PLEASE ADD THIS TO YOUR TEXT.

All in all, the same observation is repeated throughout the manuscript: you should make it clear (in the manuscript) that this is just a hint, that points to a possible AEFI, but to be sure, you should sequence NS proteins and detect no DENV4 variants (as the Qdenga construct is). Please make it clear for the reader and then it will be in a publishable state for Viruses.

My best,

The Reviewer

1)

Round 2

Reviewer 2 Report

Comments and Suggestions for Authors

I sincerely thank the authors for the clarifications. However, given the context that the detected strains were not circulating as they were vaccine sequences, and that these are present only in rare circumstances, I disagree with the message that there is a need to survey for vaccine viremia due to this case study.